# In Vitro Evaluation of Antiprotozoal Properties, Cytotoxicity Effect and Anticancer Activity of New Essential-Oil Based Phytoncide Mixtures

**DOI:** 10.3390/molecules28031395

**Published:** 2023-02-01

**Authors:** Hubert Iwiński, Henryk Różański, Natalia Pachura, Aleksandra Wojciechowska, Tomasz Gębarowski, Antoni Szumny

**Affiliations:** 1Department of Food Chemistry and Biocatalysis, Wrocław University of Environmental and Life Sciences, 50-375 Wrocław, Poland; 2AdiFeed Sp. z o.o., Opaczewska, 02-201 Warsaw, Poland; 3Laboratory of Industrial and Experimental Biology, Institute for Health and Economics, Carpathian State College in Krosno, 38-400 Krosno, Poland; 4Independent Scholar, 02-201 Warsaw, Poland; 5Department of Biostructure and Animal Physiology, Wrocław University of Environmental and Life Sciences, 50-375 Wrocław, Poland

**Keywords:** phytoncides, antiprotozoal, cytotoxicity, essential oils, LD_50_, NMR, GC-MS, antiparasitic activity, anticancer

## Abstract

Protozoa, in both humans and animals, are one of the leading causes of disease. International programmes introduced in many countries have helped reduce the incidence of disease. However, it has recently become increasingly difficult to achieve the goals set for the coming years. One of the main reasons for this, as with other pathogenic organisms, such as bacteria and fungi, is the increasing resistance to current methods of treating and preventing infection. Therefore, new therapies with high efficacy are needed. In the present study, the novel mixtures of essential oils (EOs), clove, garlic, Ceylon cinnamon, and rosemary with organic acids (acetic, propionic, lactic) and metal ions (Cu, Mn, Zn) were tested against five selected model protozoa (*Euglena gracilis*, *Gregarina blattarum*, *Amoeba proteus*, *Paramecium caudatum*, *Pentatrichomonas hominis*). The cytotoxicity and potential anticancer activity of the obtained combinations were tested on the human fibroblasts (NHDF) and human cancer cell lines (A549, MCF7, LoVo, HT29). All of the mixtures showed very good antiprotozoal properties. The most efficient were the combination of clove and rosemary essential oils, mixtures of acids, and Mn ions. The LD_50_ values were in the range of 0.001–0.006% and the LD_100_ values were 0.002–0.008%. All of the tested mixtures did not show cytotoxicity against normal cells, but did show growth inhibition against cancer cell lines. The most cytotoxic against cancer cells were combinations with cinnamon essential oil. Nevertheless, the proposed combinations containing essential oils, organic acids, and metal ions have high antiprotozoal activity, with low toxicity to healthy human cells.

## 1. Introduction

Protozoa-caused diseases are a significant problem that affect millions of people around the world, directly and indirectly. The most spread and deadliest are malaria, babesiosis, human African trypanosomiasis Chagas disease and toxoplasmosis. In 2021, malaria caused 619,000 deaths globally, 95% of which were in 29 countries [1]. COVID-19, the pandemic spread all over the world, caused a long period of isolation in many countries, and between December 2019 and January 2023, caused more than 6.7 million deaths globally [2]. Diseases caused by protozoa are also a challenge in animal husbandry, such as broiler chickens, laying hens or swine, at every stage of life. Each year, there are more than 14 billion-USD losses in the broiler chicken industry alone [3,4,5]. Moreover, some protozoa can be transmitted from animals to humans. These zoonotic diseases cause a more significant problem [5,6].

The existing methods used for protozoa control and prevention are becoming ineffective. This is due to their increasing resistance to the chemotherapeutics used [7,8]. One of the main reasons is the overuse and misuse of antibiotics, both in animal production and in medicine. Another very important element is the use of preventive chemotherapy. With preventive therapy, it is possible to control the development of a pathogen and, thus, its spread. Unfortunately, this also has negative consequences. Continuous exposure of the microorganism to chemotherapeutics can lead to a more rapid development of resistance [7]. For example, in recent years, *Plasmodium falciparum* has had a narrative resistance of *Plasmodium falciparum* to artemisinin derivatives and almost complete resistance to quinolones [9]. Today, the most effective form of treatment is the so-called ‘combination therapies’, or “combination drug”, which involves merging two or more substances to achieve a synergistic effect [10,11]. In the case of *Plasmodium falciparum*, these include artemether-lumefantrine, artesunate-mefloquine or artesunate-pyronaridine [12]. Other examples of combination drugs used against *Trypanosoma brucei* are temozolomide-eflornithine and temozolomide-melarsoprol [10]. The most common combinations are of the available and used active ingredients used for certain diseases. This is due to the potentially shorter time needed to obtain marketing authorisation for a drug containing active pharmaceutical ingredients (APIs) that are already present on the market.

Promising results have been obtained with substances of plant origin. Essential oils, alkaloids or saponins, among others, have been shown to have very potent effects [13,14,15,16].

As shown in the WHO’s data and scientific publications, such combinations are an effective alternative, but it should be noted that they can exhibit cytotoxic properties [17].

Many essential oils are currently the subject of research by scientists due to their anticancer properties. Research is being conducted not only on essential oils, but also on the compounds found in them, such as carvacrol, eugenol, linalool and citral. The analyses carried out prove the effectiveness of the use of preparations containing essential oils or active compounds of them [18,19,20,21,22]. The mode of action is not known for all essential oils. However, the properties are the result of necrosis, apoptosis, the lack of the proper functioning of cell organelles, the inhibition of angiogenesis, or cell cycle arrest. These mechanisms are primarily a consequence of their cytotoxicity and the increased permeability of the cell membrane. However, it could also be based on a change in the concentration gradient between the environment and the cell; reduced ATP production or decreased mitochondrial potential may also be responsible for their anticancer properties [23,24,25]. The importance of studying the properties and cytotoxicity of plant-derived products is shown by the example of *Acorus calamus*. In the 1980s, the Food and Drug Administration (FDA) banned the use of calamus in food due to its β-azarone content and potential carcinogenic effects [26]. However, as studies on calamus products and β-azarone show, it may have just the opposite properties and exhibit anticancer effects [27,28,29,30]. Furthermore, the combination of essential oils with metal ions or current anticancer therapies may prove to be a much better solution, not only because it is more effective, but also because it reduces the negative consequences of the current treatment or its cytotoxicity towards healthy cells [23,31,32].

The aim of the present study was to determine the activity of mixtures containing essential oil (clove (*Syzygium aromaticum* (L.) Merr. and Perry), garlic (*Allium sativum* L.), cinnamon (*Cinnamomum verum* J. Presl) or rosemary (*Rosmarinus officinalis* L.)) with metal ions (Zn, Cu and Mn) and organic acids (acetic, propionic and lactic). Due to the fact that the proposed phytoncidal mixtures could be used in practice, e.g., in animal breeding, we decided to carry out, in addition to the main antiprotozoal activity, an evaluation of their safety and cytotoxic activity. The proposed mixtures are novel, and there is no indication of any predictable effect on cell lines. Potential cytotoxic activity against healthy human cell lines would be limiting to their practical use. We chose cytotoxicity assays toward human epithelial fibroblasts and several human cancer cell lines as a widely used model.

## 2. Results

### 2.1. Activity against Selected Protozoa

The LD_50_ and LD_100_ values of 48 different combinations were determined against the selected protozoa. Each composition is shown in the Table 1. Two common antibiotics, chloramphenicol and metronidazole, were used as reference substances. The essential oils used in the study, without organic acid or mineral salts, showed very promising results in some of the analyses, comparable to the effectiveness of antibiotics. The most promising and strongest antiprotozoal activity was obtained for clove oil. However, the other essential oils also showed potential for further research.

The mixtures containing an essential oil, a single acid and a metal ion were characterized by significantly better antiprotozoal efficacy than essential oils alone. As can be observed, the effective antiprotozoal concentration was much lower than LD_50_ and LD_100_ for the pure essential oils. What is more important, they were lower than LD_50_ and LD_100_ values obtained for the antibiotics used as a reference substances. This fact can be observed regardless of the essential oil. In compositions containing essential oil, metal ions and a single organic acid, as in the case of a single essential oil, the strongest antiprotozoal properties were shown by all of the combinations containing clove essential oil. For LD_50_, it was 0.01–0.02%, while LD_100_ was 0.015–0.03%. However, for the other essential oils and their combinations with acids, the LD_50_ and LD_100_ values were 0.01–0.05% and 0.02–0.07%, respectively. Combinations containing garlic, cinnamon and rosemary oils, in contrast to the clove essential oil mixtures, showed relatively weak antiprotozoal activity against *Pentatrichomonas hominis*. The best antiprotozoal properties against all of the tested organisms were observed for combinations with lactic acid and Mn ions, for all of the essential oils used.

However, regardless of the essential oil used, the best results were obtained for the composition containing a mixture of organic acids. The combinations that presented the most powerful antiprotozoal properties were those that included clove essential oil. The LD_50_ and LD_100_ values for the SMMn and SMZn were in the range of 0.001–0.003% and 0.002–0.005%, respectively. The SMCu mixture shows slightly weaker properties, with LD_50_: 0.001–0.004% and LD_100_: 0.003–0.006%. Very good antiprotozoal activity was shown in all three combinations with rosemary essential oil (RMCu, RMMn and RMZn). The LD_50_ values were very promising (0.001–0.003%); however, further tests showed that to reach LD_100_ values, much more concentrated solutions were needed (0.004–0.009%). The remaining essential oils obtained results that were not as promising as the previous two; nevertheless, they also showed very good antiprotozoal activity, higher than that of the antibiotics currently used. Two combinations containing manganese ions (GMMn and CMMn) showed the lowest LD_50_ and LD_100_ values for garlic and cinnamon essential oils. All of the results are presented in Table 2, Table 3, Table 4, Table 5, Table 6, Table 7, Table 8 and Table 9.

### 2.2. Compositions Analysis (GC-MS)

Table 10 and Table 11 show that the compositions contained typical compounds for clove and garlic essential oils compared to the retention indices from the Adams and NIST databases. Table 12 presents the results of NMR (^1^H and ^13^C) analyses. Composition of Ceylon cinnamon and rosemary essential oils are presented in Table 13 and Table 14.

#### 2.2.1. Clove Essential Oil

Two of the essential oils used, clove and cinnamon, were characterized by a relatively high dominance of one main compound in the composition. In the case of the clove bud essential oil, eugenol accounted for more than 70% of the content. This composition was complemented by caryophyllene (14%) eugenyl acetate (9.1%) and humulene (3.4%).

**Table 10 molecules-28-01395-t010:** GC-MS profile of clove essential oil.

No.	Peak Name	KI Exp. ^1^	KI Adams ^2^	KI NIST ^3^	CAS ^4^	Content [%] ^5^	Identification
1	Chavicol	1258		1255	501-92-8	0.03	S, MS, KI
2	Eugenol	1364	1359	1357	97-53-0	71.45	S, MS, KI
3	α-Copaene	1379	1375	1376	3856-25-5	0.10	MS, KI
4	*trans*-Caryophyllene	1423	1419	1419	13877-93-5	14.08	S, MS, KI
5	α-Humulene	1457	1454	1454	6753-98-6	3.45	S, MS, KI
6	Zonarene	1527	1529	1527	41929-05-9	0.53	MS, KI
7	Eugenyl acetate	1532		1524	93-28-7	9.11	S, MS, KI
8	Unknown	1557				0.29	-
9	Caryophyllene oxide	1587	1583	1581	1139-30-6	0.96	S, MS, KI

^1^ Experimental retention indices calculated against n-alkanes. ^2^ Retention indices according to the Adams database. ^3^ Retention indices according to the NIST20 database. ^4^ Chemical Abstracts Service. ^5^ % calculated from Total Ion Chromatogram (TIC). The MS spectrum is presented as S1; Identification based on: S—standard compound available; MS—mass spectrum; KI—Kovatc indices.

#### 2.2.2. Garlic Essential Oil

The garlic essential oil did not contain a single main compound and its composition of individual molecules are relatively evenly distributed among several of them. It contained the vast majority of allicin derivatives. These were, respectively, diallyl disulphide (30.7%), diallyl trisulfide (25%) and diallyl tetrasulfide (14.5%).

Although there are many publications that describe the composition of essential oils extracted from garlic by gas chromatography, there are many doubts about the actual accuracy of such a measurement [33]. The thermal degradation of diallyl disulphides at about 150 °C (CG condition) to mono-, tri-, and terta-disulphides, as well as their rearrangement to heterocyclic thiopyranes, trithiolane or tetrathianes, has been demonstrated [34,35]. We decided to carry out a comparative analysis of garlic EO, using the mildest method, which was nuclear magnetic resonance spectroscopy (NMR). This technique does not require exposure to high temperatures and allows for the unambiguous determination of the presence of mono- and ooligothio-derivatives of diallyl compounds. The methylene group of the allyl fragment was very well separated on the ^1^H NMR spectrum in the 3.1 to 3.65 ppm region. The ^13^C, as well as the correlative spectra, confirmed the unequivocal identification of the compounds. We obtained different values from those presented in Table 12 for the proportion of diallyl derivatives. As a predominate, we found diallyl disulphide, which was presented in nearly 50% of the EO. Tri- and hexa-allyl disulfides were the next compounds present in the mix, which were higher than 10%. Contrary to the GC-MS profile, diallyl tetrasulfide was detected bellow the quantification limit. The share of the latter compound in the chromatographic analysis was as high as 14%. As there are no reports describing the chemical shifts of diallyl heptasulphide, we assumed the presence of this compound in the EO. Additionally, there are no EI-MS spectra or RI values in the NIST databases. NMR spectra’s are presented at Figure 1 and Figure 2. 

**Table 11 molecules-28-01395-t011:** GC-MS profile of garlic essential oil.

No.	Peak Name	KI Exp. ^1^	KI NIST ^2^	CAS ^3^	Content [%] ^4^
1	Diallyl sulfide	860	861	592-88-1	2.62
2	Disulfide, methyl 2-propenyl	923	920	2179-58-0	0.6
3	3H-1,2-Dithiole	949	952	288-26-6	0.9
4	Diallyl disulphide	1081	1081	2444-49-7	35.2
5	Trisulfide, methyl 2-propenyl	1140	1142	34135-85-8	1.18
6	4-Methyl-1,2,3-trithiolane	1154	1154	116664-29-0	6.21
7	4H-1,2,3-Trithiine	1202	1202	290-30-2	0.25
8	Trisulfide, di-2-propenyl	1300	1297	2050-87-5	28.75
9	5-Methyl-1,2,3,4-tetrathiane	1364	1364	116664-30-3	3.72
10	1-(1-(Methylthio)propyl)-2-propyldisulfane	1440	1431	126876-22-0	0.53
11	Diallyl tetrasulfide	1542	1540	2444-49-7	16.59
12	Disulfide, 1-(1-propenylthio)propyl propyl	1585	1592	143193-11-7	0.66
13	1-Allyl-2-(1-(allylthio)propan-2-yl)disulfane	1594	1597	116664-22-3	1.54
14	1,2,3,5-Tetrathiane, 4,6-diethyl-, trans-	1641	1640	137363-93-0	1.25

^1^ Experimental retention indices calculated against n-alkanes. ^2^ Retention indices according to the NIST20 database. ^3^ Chemical Abstracts Service. ^4^ % calculated from Total Ion Chromatogram (TIC). All compounds were identified on the basis of mass spectra.

**Table 12 molecules-28-01395-t012:** Composition of garlic EO according to ^1^H analysis.

	Precentage ^a^	^1^ H Multiplet (ppm)	^13^ C (ppm)	Reference
DA monosulphide	8.8	3.12 (dt, *J* = 7.1, 1.1 Hz, 2H)	33.35	[34]
DA disulphide	49.0	3.36 (dt, *J* = 7.4, 1.1 Hz, 2H)	42.33	[35]
DA trisulphide	24.5	3.53 (dt, *J* = 7.3, 1.1 Hz, 2H)	42.12	[35]
DA tetrasulphide	- ^b^	3.58 (d, *J*= 7.2 Hz, 2H)	-	[35]
DA pentasulphide	2.9	3.36 (dt, *J* = 7.4, 1.1 Hz, 2H)	42.50	[35]
DA hexasulphide	11.8	3.61 (dt, *J* = 7.3, 1.0 Hz, 2H)	42.47	[35]
DA heptasulphide ^c^	2.9	3.64 (dt, *J* = 7.3, 1.1 Hz, 2H)	42.62	-

^a^ according to ^1^H methylene group integration; ^b^ bellow limit of quantification; ^c^ tentatively identified.

**Figure 1 molecules-28-01395-f001:**
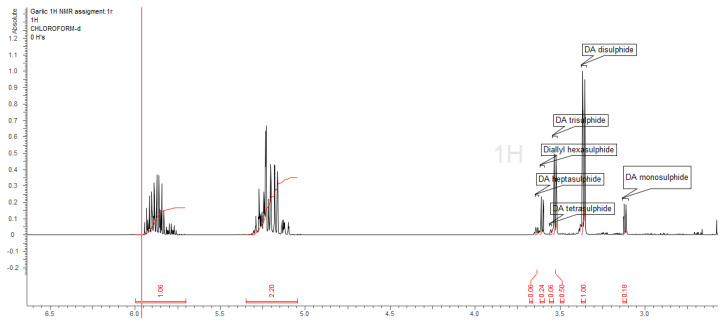
Garlic essential oil ^1^H NMR spectrum.

**Figure 2 molecules-28-01395-f002:**
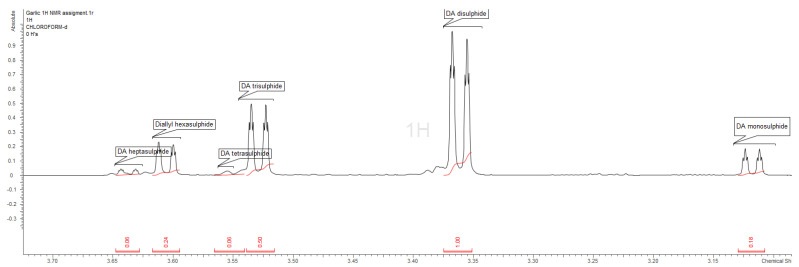
Fragment of garlic essential oil ^1^H NMR spectrum.

#### 2.2.3. Ceylon Cinnamon Essential Oil

In the case of the second essential oil, cinnamon, its main component was *trans*-cinnamaldehyde (54.7%). Other compounds with the highest contribution were linalool (5.5%), cinnamyl acetate (4.3%) and *trans*-caryophyllene (4.1%).

**Table 13 molecules-28-01395-t013:** GC-MS profile of Ceylon cinnamon essential oil.

No.	Peak Name	KI Exp. ^1^	KI Adams ^2^	KI NIST ^3^	CAS ^4^	Content [%] ^5^	Identification
1	α-Pinene	939	939	937	80-56-8	1.44	S, MS, KI
2	Benzaldehyde	966	960	962	100-52-7	0.88	S, MS, KI
3	β-Pinene	980	979	979	127-91-3	0.68	S, MS, KI
4	*p*-Cymene	1028	1024	1025	99-87-6	2.23	S, MS, KI
5	Limonene	1032	1029	1030	138-86-3	0.68	S, MS, KI
6	Eucalyptol	1035	1031	1034	470-82-6	3.49	S, MS, KI
7	γ-Terpinene	1063	1059	1060	99-85-4	1.83	S, MS, KI
8	Terpinolene	1090	1088	1088	586-62-9	0.87	S, MS, KI
9	Linalool	1100	1090	1099	78-70-6	5.50	S, MS, KI
10	Phenethyl alcohol	1116		1116	60-12-8	1.26	S, MS, KI
11	Terpinen-4-ol	1180	1177	1177	562-74-3	0.71	S, MS, KI
12	α-Terpineol	1192	1188	1189	98-55-5	2.94	S, MS, KI
13	γ-Terpineol	1198	1199	1197	586-81-2	0.21	S, MS, KI
14	*cis*-Cinnamaldehyde	1224	1219	1219	57194-69-1	0.22	S, MS, KI
15	Geraniol	1259	1252	1255	106-24-1	0.58	S, MS, KI
16	2-Phenethyl acetate	1261		1258	103-45-7	1.42	S, MS, KI
17	*trans*-Cinnamaldehyde	1277	1270	1270	14371-10-9	54.70	S, MS, KI
18	*trans*-Anethole	1289	1284	1284	104-46-1	2.44	S, MS, KI
19	*trans*-Cinnamyl alcohol	1308	1304	1312	4407-36-7	0.44	S, MS, KI
20	Limonenal	1329	1326		6784-13-0	0.16	MS, KI
21	α-Terpinyl acetate	1353	1349	1350	80-26-2	1.66	S, MS, KI
22	α-Longipinene	1355	1352	1353	5989-08-2	0.15	MS, KI
23	Eugenol	1362	1359	1357	97-53-0	3.20	S, MS, KI
24	α-Copaene	1379	1376	1376	3856-25-5	1.37	S, MS, KI
25	*trans*-Caryophyllene	1422	1419	1419	87-44-5	4.14	S, MS, KI
26	α-*trans*-Bergamotene	1439	1432	1435	13474-59-4	0.57	S, MS, KI
27	*trans*-Cinnamyl acetate	1449	1446	1446	21040-45-9	4.27	MS, KI
28	α-Humulene	1457	1454	1454	6753-98-6	0.23	S, MS, KI
29	Chavibetol acetate	1532	1525		61499-22-7	0.17	MS, KI
30	γ-*trans*-Bisabolene	1546	1531	1533	70286-32-7	0.24	MS, KI
31	Caryophyllene oxide	1587	1583	1581	1139-30-6	0.18	S, MS, KI
32	Benzyl benzoate	1769	1760	1762	120-51-4	1.14	S, MS, KI

^1^ Experimental retention indices calculated against n-alkanes. ^2^ Retention indices according to the Adams database. ^3^ Retention indices according to the NIST20 database. ^4^ Chemical Abstracts Service. ^5^ % calculated from Total Ion Chromatogram (TIC). Identification based on: S—standard compound available; MS—mass spectrum; KI—Kovatc indices.

#### 2.2.4. Rosemary Essential Oil

The rosemary essential oil was characterized by the presence of four main compounds, accounting for more than 70% of its composition. Unfortunately, the relative proximity of the peaks resulting from the KI values (1032 and 1035) for limonene and eucalyptol, respectively, prevented their effective separation and verification of their intrinsic validity. However, collectively, they account for 47.9% of the composition of the essential oil of rosemary. Other constituents included in the composition of this essential oil were α-pinene (13.1%) and camphor (11.75%).

**Table 14 molecules-28-01395-t014:** GC-MS profile of rosemary essential oil.

No.	Peak Name	KI Exp. ^1^	KI Adams ^2^	KI NIST ^3^	CAS ^4^	Content [%] ^5^	Identification
1	Tricyclene	926		925	508-32-7	0.13	S, MS, KI
2	α-Thujene	932	930	929	2867-05-2	0.15	S, MS, KI
3	α-Pinene	939	939	937	80-56-8	13.09	S, MS, KI
4	Camphene	954	954	952	79-92-5	5.09	S, MS, KI
5	Benzaldehyde	960	960	962	100-52-7	0.00	S, MS, KI
6	Sabinene	977	975	974	3387-41-5	0.01	S, MS, KI
7	β-Pinene	980	979	979	127-91-3	6.51	S, MS, KI
8	β-Myrcene	993	990	991	123-35-3	1.07	S, MS, KI
9	α-Fellandrene	1006	1002	1005	99-83-2	0.17	S, MS, KI
10	3-Carene	1013	1011	1011	13466-78-9	0.03	S, MS, KI
11	α-Terpinene	1020	1017	1017	99-86-5	0.47	S, MS, KI
12	*p*-Cymene	1028	1024	1025	99-87-6	2.31	S, MS, KI
13	Limonene + Eucalyptol	1036	1029/1031	1030/1032	138-86-3	47.93	S, MS, KI
14	γ-Terpinene	1063	1059	1060	99-85-4	0.52	S, MS, KI
15	*trans*-Sabinene hydrate	1071	1070	1070	17699-16-0	0.04	MS, KI
16	Terpinolene	1091	1088	1088	586-62-9	0.30	S, MS, KI
17	Linalool	1100	1096	1099	78-70-6	0.82	S, MS, KI
18	Fenchol	1116	1116	1113	1632-73-1	0.04	S, MS, KI
19	*trans*-Sabinol	1143	1142	1142	471-16-9	0.05	MS, KI
20	Camphor	1148	1146	1144	464-49-3	11.75	S, MS, KI
21	Isoborneol	1160	1160	1157	124-76-5	0.17	MS, KI
22	endo-Borneol	1169	1169	1167	507-70-0	2.55	MS, KI
23	Terpinen-4-ol	1180	1177	1177	562-74-3	0.40	S, MS, KI
24	α-Terpineol	1192	1188	1189	98-55-5	1.22	S, MS, KI
25	Verbenone	1211	1205	1205	80-57-9	0.11	S, MS, KI
26	Bornyl acetate	1287	1285	1285	76-49-3	0.63	S, MS, KI
27	α-Copaene	1380	1375	1376	3856-25-5	0.27	MS, KI
28	β-*trans*-Caryophyllene	1424	1417	1419	87-44-5	3.30	S, MS, KI
29	Aromandendrene	1444	1441	1440	489-39-4	0.05	MS, KI
30	Humulene	1458	1454	1454	6753-98-6	0.26	S, MS, KI
31	γ-Muurolene	1480	1479	1477	30021-74-0	0.20	MS, KI
32	γ-Cadinene	1517	1513	1513	39029-41-9	0.12	MS, KI
33	δ-Cadinene	1526	1523	1524	483-76-1	0.25	MS, KI

^1^ Experimental retention indices calculated against n-alkanes. ^2^ Retention indices according to the Adams database. ^3^ Retention indices according to the NIST20 database. ^4^ Chemical Abstracts Service. ^5^ % calculated from Total Ion Chromatogram (TIC). Identification based on: S—standard compound available; MS—mass spectrum; KI—Kovatc indices.

### 2.3. Evaluation of Biological Activity In Vitro

The assessment of the impact on cell cultures was performed using the sulforhodamine B (SRB) assay. This test is a good tool for preliminary studies. From the results of the test, both the cytotoxic and cytostatic effects of the tested compounds can be determined. The test measures both the effects on the basis of the amount of protein, and the result is not influenced by the inhibition or generation of free radicals by the compounds. The compounds tested are oils. Cancer cells, including those used in the tests, have a simplified morphological structure and physiology adapted to rapid cell division. The test compounds, oils, are, due to their nature, easily penetrated into the cells. Tumors cells, by virtue of their physiology, are unable to metabolize such compounds. This usually results in the inhibition of cell growth and, to a lesser extent, apoptosis or cytotoxicity.

As a result of the study, no cytotoxicity of the tested compounds was found in the tested concentration range. A decrease in the amount of protein was observed in the test cultures below the amount at the beginning of the test. Such effects are seen in the case of cytostics or other compounds with strong toxicity. Example photographs of cell cultures are included in the Appendix A.

Table 15, Table 16, Table 17 and Table 18 show the IC_50_ values for the four oils tested, together with their modifications, for normal fibroblast (NHDF) cultures and the selected tumor lines. The results obtained indicate a cytostatic effect of the tested oils. The tests showed a higher sensitivity of the tumor cells compared to the fibroblast cultures. The greatest differences in activity, seen in Table 15, are for cinnamon oil. The weakest activity was found for rosemary oil. In the case of this oil, the use of additives achieved the inhibition of cancer cell growth (Table 18). The garlic (Table 16) and clove (Table 17) oils also showed strong effects on cancer cells and weaker effects on normal cells.

## 3. Discussion

The presented tests, with combinations containing an essential oil (clove, Ceylon cinnamon, garlic or rosemary), organic acid or its mixture (acetic, propionic and lactic acid) and metal ions (Cu, Mn or Zn), revealed antiprotozoal properties. The idea of preparing a complexed mixture was based on our previous research and test. Many scientific reports demonstrate the antiprotozoal activity of essential oils or even mixtures containing two constituents, but there is a lack of data regarding compositions containing essential oils, organic acid and metal ions. [36,37,38,39].

The combinations with clove essential oil had the best antiprotozoal properties of all the mixtures used. The clove essential oil also has very good antiprotozoal properties [40]. The obtained LD_50_ and LD_100_ values were between 0.1–0.3%. Santoro et al., observed significantly better properties against the various stages of the development of *Trypanosoma cruzi* than with eugenol alone [39]. A similar situation occurred in our study, where more unexpected values were reached for the combinations with organic acids and metal ions. The LD_50_ and LD_100_ were 0.001–0.04% and 0.001–0.005%, respectively. This suggests a synergistic effect between the components used. The GC-MS analysis proved that the clove essential oil used was derived from buds. This is evidenced by the content of eugenol (71.4%), *trans*-caryophyllene (14%) and, most importantly, the presence of eugenyl acetate (9.1%) [41,42,43].

Other very promising results were obtained for the Ceylon cinnamon essential oil. The pure essential oil was very effective. However, further tests, with combinations containing acids and metal ions, were not as exceptional as had been expected. The GC-MS analyses showed the content of the main compound, *trans*-cinnamaldehyde, was 54.7%. This is a relatively low content of the head compound of cinnamon essential oil, which is in the range 65–75% [44,45,46]. In the case of the tested sample, the results were comparable with those presented by Martiniaková et al. The trans-cinnamaldehyde content was similar, but the other compounds were present in comparable amounts [47].

The mixtures containing garlic essential oils showed good results. The best antiprotozoal results for the combinations with this oil were obtained against *Amoeba proteus*. The same results and conclusions were suggested by Behnia et al. [48]; in their studies against *Entamoeba histolica*, they demonstrated the antiamoeba properties of different extracts and garlic essential oil. The activity was time-dependent and the exposure for the solutions was 24 h and 48 h. These findings correspond to the results presented in our studies. Further tests showed that combinations with garlic essential oil had good antiprotozoal properties, but the lowest from the analyzed samples.

At the initial stage if the study, the rosemary essential oil did not show the best antiprotozoal properties of the tested essential oils. The values of LD_50_ and LD_100_ were at the level of 0.1–0.3% and 0.4–0.6%, respectively. The situation was similar for its combination with single acids. However, the combination with a mixture of acids proved to be very effective against all of the protozoa tested, reaching LD_50_: 0.001–0.007% and LD_100_: 0.003–0.009% values.

At present, many literature reports can be found on combining various types of compounds and essential oils with metal ions. In these studies, a definite enhancement of their effect can be observed [32,49]. The following results confirm that the use of the addition of metal ions and organic acids increases the antiprotozoal properties of the essential oils.

One of the most serious problems arising from the use of new substances, whether in pharmaceuticals, medicine or human and animal nutrition, is the lack of data on the toxicology of new compounds, their composition or metabolites. New formulations with very promising results have often not been used in a wider context due to side effects and toxicity. The situation is similar in the case of essential oils, where sufficiently high concentrations can cause adverse reactions or toxic effects. For example, clove essential oil may act as an irritant. On the other hand, some essential oils and their main constituents, such as *Rosmarinus officinalis* (camphor), *Eucalyptus globulus* (1,8-cineole) and *Mentha* species (menthone and menthol), and, in particular, *Mentha pulegium* (pulegone), may cause hepatic damage, convulsions and hallucinations. They are also forbidden in pregnancy due to their abortifacient effect [50,51]. In our study, the toxic effect of the prepared mixtures was tested on several cell lines and cancer cell lines, for example, NHDF (normal human dermal fibroblasts), A549 (lung cancer) or MCF7 (human breast cancer). All of the essential oils used in this study have been reported as potential anticancer agents [32,52,53,54,55].

The results indicate the low toxicity of the tested oils towards normal cells and the inhibition of cancer cell growth. There were no results indicating strong cytotoxicity of the tested compounds, the main effect being the inhibition of cell growth. This confirms the safety of the tested oils. The results obtained may be indicative of their chemopreventive properties. Due to their volatile properties, the tested oils can easily work in the respiratory tract or digestive tract when applied indoors, in the same way as with pollutants reaching our body. The results obtained show a selective action of the oils against cancer cells, with the best effect seen for cinnamon oil (4–12 × stronger effect). Modification, through the addition of metals to the oils, also resulted in an improved anticancer effect. The addition of Zn significantly improved the activity of cinnamon oil (1.7–2.2 × potent, depending on the line tested) and is also important for its protective effect against upper respiratory tract infections.

All of the obtained results are very promising in terms of either their antiprotozoal activity or low cytotoxicity and their possession of good anticancer properties. Low cytotoxicity is the first step to determine the safety of the combinations for human and animals. Unfortunately, heavy metals can accumulate in tissues and organs and can cause several disorders. Moreover, some essential oils can cause allergic reactions and organic acids can cause skin and mucosal irritation. The wider use of the mixtures, for diseases caused by protozoa or cancer treatments, requires more tests and further investigation, e.g., in vivo trials. In vivo trials performed on animals, which would include intentional infection, require special ethic approval of the Local Ethics Committee.

## 4. Materials and Methods

### 4.1. Maintenance of Parasite Cultures and Evaluation of Antiprotozoal Activity

For the further tests, five different protozoa were chosen. They belong to the same taxonomic groups as the most widespread pathogenic protozoa. Three of them, *Amoeba proteus*, *Paramecium caudatum* and *Euglena gracilis*, represent aquatic protozoa, while *Gregarina blattarum*, *Pentatrichomonas hominis* live in the digestive track of cockroaches and humans, respectively.

The aquatic protozoa were isolated in Krosno from the freshwater river (river Badoń, 49°39′59.8″ N 21°46′28.1″ E, Krosno, Subcarpathian Voivodeship, Poland). The identification of the protozoa was performed on the descriptions and drawings of W. A. Dogiel [56] and J. Hempel-Zawitkowska [57].

All of the analyzed protozoa were cultivated under different conditions. Hay infusion was the medium for the cultivation of *Paramecium* [58,59,60]. For *Euglena*, the solution was used according to Wu et al. [61]. *Pentatrichomonas hominis* was kept in solution according to Chomicz et al. after it was isolated from stool samples [62]. *Tetrahymena* and *Chilomonas* were ciliates, used as an example feed for *Amoeba proteus*, and were cultivated in Prescott medium [63,64]. Only one protozoa was not cultivated and was isolated from the cockroaches. Gregarines, after isolation, proposed by J. Moraczewski [65], were placed in Ringer solution on a watch glass.

To determine the LD_50_ and LD_100_ values, different combinations and their concentrations were tested. Each time, four-fold replicates and blank were used. The Reed-Muench method was used for determining the LD_50_ and LD_100_ values. The protozoa were treated with different concentrations of the samples and observed on a watch glass, for 3 to 5 min.

The resulting phytoncide-metal mixture and phytoncide alone were dissolved in an aqueous solution of polysorbate 80 (0.05%) and applied to the watch glass. No biocidal activity of polysorbate 80 was observed at these concentrations. Chloramphenicol and metronidazole were used as standard substances for the control of protozoa. Antibiotics were used at a concentration of 5 mg/mL and diluted from stock solutions to achieve LD_50_ and LD_100_.

### 4.2. Essential Oils

The essential oils were purchased from three companies. The cinnamon essential oil was provided by Food Base Kft. (Gödöllő, Hungary); the garlic essential oil from Synthite Industries Pvt., Ltd. (Kolenchery, Kerala, India); the rosemary and clove essential oil from De Monchy Aromatics Ltd. (Poole, Dorset, UK).

### 4.3. Chemicals and Reagents

The organic acids (acetic acid 99%, propionic acid 99.5% and lactic acid 85%) and other chemical reagents were purchased from Sigma-Aldrich (St. Louis, MO, USA) and comply with FCC and FG standards. The standards for chromatographical analyses were bought in Sigma-Aldrich (St. Louis, MO, USA), UQF (Wrocław, Poland) and Metasci (Toronto, ON, Canada), as well as our own collection of chemicals in the Department of Food Chemistry and Biocatalysis. The purity and percentage composition, according to the supplier’s specification, was minimum ≥95%.

### 4.4. Phytoncides Mixture Preparation

Organic acids and essential oils were mixed in the same amount (100 mL). The mixture of organic acids was prepared in ratio 1:1:1. The composition was mixed, and mineral salts were added in the amount of 5 g. The salts were: manganese (II) chloride (2.18 g of ions Mn^2+^), copper (II) carbonate hydroxide (2.87 g of ions Cu^2+^) and zinc carbonate (2.61 g of ions Zn^2+^). The composition was heated and then left to cool overnight. After that time, the solution of one, two or three phases was filtered through paper filter. The composition was diluted to prepare a solution in the range of 0.001% to 1.5%.

### 4.5. GC-MS Analysis

The profile of the essential oils investigated was evaluated using the GC-MS technique, according to the protocol [66]. The identification of all of the volatile components was based on a comparison of the mass spectra with the mass spectra of the compound obtained experimentally, available in the NIST20 database. Additionally, the retention indices (RI), obtained experimentally, were calculated using macro [67] and were compared with the RI available in the NIST20 database and the data from the literature [68]. GCMS Post-run analysis software version 4.45 (Shimadzu Company, Kyoto, Japan) and ACD/Spectrus Processor (Advanced Chemistry Development, Inc., Toronto, ON, Canada) were used to process the data. The quantification of the identified constituents was performed by calculation based on the amount of added internal standard and expressed as a percentage of the integrated peaks’ area. Analysis was performed using the Shimadzu 2020 apparatus (Varian, Walnut Creek, CA, USA) equipped with a Zebron ZB-5 MSI column (30 m × 0.25 mm × 0.25 μm) column (Phenomenex, Torrance, CA, USA). The temperature of the GC oven was programmed from 50 °C to 250 °C at a rate of 3.0 °C and kept for 3 min. Scanning was performed from 35 to 550 m/z in electronic impact (EI) at 70 eV and ion source temperature 250 °C. Samples were injected at split ratio 1:10 and gas helium was used as the carrier gas at a flow rate of 1.0 mL/min. Garlic and rosemary Eos were analyzed on Varian CP-3800/Saturn 2000 apparatus (Varian, Walnut Creek, CA, USA) and compounds were separated by on Zebron ZB-5MSi (30 m × 0.25 mm × 0.25 µm) column. GC temperature program: initially 50 °C, then to 180 °C at 4.0 °C/min ratio, and finally to 250 °C at 10 °C/min ratio. As a carrier gas helium with linear velocity 35.0 cm/s; split ratio 1:10 was used. MS operational conditions: ion source temperature 250 °C; electron impact (EI) ionization at 70 eV; scanning range between 35 and 300 m/z.

### 4.6. NMR Measurement

The ^1^H NMR and ^13^C spectra of EOs were recorded in a CDCl_3_ solution on a Bruker Avance™ 600 MHz spectrometer (Bruker, Billerica, MA, USA). Two different measurements, that is 25 µL (for ^1^H measurement) or 250 µL for ^13^C and correlative spectra), of essential oil were dissolved in 600 µL of CDCl_3_ to record the spectra. The data were processed on the ACD Spectrus Processor 2021.2.1, Advanced Chemistry Development, Inc. Toronto, ON, Canada.

### 4.7. Cell Culture

Normal cell lines–dermal fibroblasts (NHDF) purchased from LONZA (Verviers, Belgium) and cancer cell lines A549 (lung cancer), MCF7 (breast cancer) and LOVO and HT29 (colorectal adenocarcinomas) were used in the study. Tumour lines were purchased from the European Collection of Authenticated Cell Cultures (ECACC). The cells were cultured under standard 37 °C and 5% CO_2_ conditions. The cell lines were thawed for a minimum of 2 weeks prior to the start of the study and passaged after reaching full confluence. The cells were cultured in medium supplemented with 10% FBS and appropriate NHDF, HT29-DMEM, A549 and MCF7-EMEM, LOVO-DMEM/F12 media. The media were supplemented with antibiotics and L-glutamine. All of the reagents were purchased from Biological Industries—now part of Sartorius (Kibbutz Beit Haemek, Israel). Materials for culture bottles, plates, tubes were purchased from SPL Life Sciences (Pocheon, Korea).

### 4.8. Evaluation of Biological Activity on Cell Cultures

The activity assessment was performed according to the National Cancer Institute guidelines for screening human tumour lines and based on the basis of our own studies [69]. Tumour cell lines and normal human cells cultured in a suitable medium containing 5% fetal serum were used for the study. To prepare the cells for the experiment, the medium was harvested and inactivated with trypsin. Trypsin has the effect of detaching cells from the medium. A portion of the cells were then collected into tubes and trypsin inactivated with the medium. The quality and viability of the cells used in the study was measured using a NucleoCounter^®^ NC-200 reader (Chemometec, Denmark).vCell viability is measured using dedicated cassettes containing acridine orange (AO) and 4′,6-diamidino-2-phenylindole (DAPI).

The cells were counted in line NC200 (Chemometek, Allerod, Denmark) and the final cell count after inoculation in a 96-well plate was approximately 1 million cells (1 × 10^4^ cells per well). After inoculation, the plates were incubated at 37 °C, 5% CO_2_ and 100% relative humidity for 24 h before phytoncides were added. After 24 h of incubation, one plate from each cell line was fixed with 50% (*w*/*v*) TCA to represent cell population measurements for each tumour line at the time of oil addition (T0). The oils were prepared at a concentration of 10 mg/mL and the volume added to each microtiter well was 1 mg/mL, 0.5 mg/mL and 0.1 mg/mL, respectively. The plates were then incubated for an additional 48 h at 37 °C, 5% CO_2_, 95% air and 100% relative humidity. After this time, the contents of the wells were fixed by adding 30 μL of 50% (*w*/*v*) TCA and incubated for 60 min at 4 °C. The supernatant was discarded and the plates were washed five times with tap water and air-dried. To each well, 100 µL of a 0.4% (*w*/*v*) solution of sulforhodamine B (SRB) in 1% acetic acid was added and plates were incubated for 10 min at room temperature. After staining, the unbound dye was removed by washing five times with 1% acetic acid and the plates were air-dried. Doxorubicin was used as the positive control, in final concentration 10 µM. The bound dye was dissolved in 10 mM Trisma Base and the absorbance was read on a MultiscanGo reader (Thermo Scientific, Waltham, MA, USA) at 515 nm.

The absorption results obtained were compared with those of the control and the T0 control. In the case of absorbance values below the T0 control, such a result provided information on the cytotoxic properties of the tested compounds. If the results were between T0 and the control value, the result indicated an inhibitory effect on cell growth.

## 5. Conclusions

The proposed new mixtures containing essential oils, organic acids and metal ions have not yet been obtained and described in the scientific literature. Those combinations containing three different active substances showed very high antiprotozoal efficacy at very low concentrations. The most effective combinations turned out to be solutions containing, in addition to essential oil and metal ions, a mixture of organic acids. The most effective combinations against the analyzed protozoa were those containing clove essential oil. Very similar results were also obtained for rosemary essential oil. A slightly lower effectiveness was characterized by mixtures with garlic and ceylon cinnamon essential oils. However, it should be noted that even the least effective combinations, in most cases, were significantly more effective than the reference substances, chloramphenicol and metronidazole. Moreover, the analyses of the cytotoxic effect against human cancer cell lines showed a very promising effect. The tests performed on normal human cell lines showed low toxic effects. However, on the other hand, the tested compositions inhibited the growth of cancer cells. The highest anticancer, or more precisely, chemopreventive properties, were obtained for the cinnamon essential oil mixtures and combinations containing zinc ions.

The collected data allow us to conclude that the discovered combinations, under in vitro conditions, have very good antiprotozoal properties and, importantly, they have low toxicity against healthy human cell lines and inhibit the growth of cancer cell lines. Further studies should be conducted to determine their potential side effects, metabolism and accumulation in tissues under in vivo conditions.

## Figures and Tables

**Table 1 molecules-28-01395-t001:** Combinations obtained during the research.

Essential Oil	Acetic Acid (A)			Propionic Acid (P)			Lactic Acid (L)			Mixture of Acids (M)		
	Cu	Mn	Zn	Cu	Mn	Zn	Cu	Mn	Zn	Cu	Mn	Zn
Clove (*Syzygium aromaticum* (L.) Merr. and Perry) (S)	SACu	SAMn	SAZn	SPCu	SPMn	SPZn	SLCu	SLMn	SLZn	SMCu	SMMn	SMZn
Garlic (*Allium sativum* L.) (G)	GACu	GAMn	GAZn	GPCu	GPMn	GPZn	GLCu	GLMn	GLZn	GMCu	GMMn	GMZn
Ceylon cinnamon (*Cinnamomum verum* J. Presl) (C)	CACu	CAMn	CAZn	CPCu	CPMn	CPZn	CLCu	CLMn	CLZn	CMCu	CMMn	CMZn
Rosemary (*Rosmarinus officinalis* L.) (R)	RACu	RAMn	RAZn	RPCu	RPMn	RPZn	RLCu	RLMn	RLZn	RMCu	RMMn	RMZn

**Table 2 molecules-28-01395-t002:** LD_50_, LD_100_ values [%] of clove essential oil (*Syzygium aromaticum* (L.) Merr. and Perry) and the components used in the study.

Protozoa	CH ^a^	M ^b^	Acetic Acid	Propionic Acid	Lactic Acid	Mixture of Acids ^c^	MnCl_2_ Solution ^d^	CH_2_Cu_2_O_5_ Solution ^e^	ZnCO_3_ Solution ^f^	Catalyst Solution ^g^	Clove Essential Oil (*Syzygium aromaticum* (L.) Merr. and Perry)
*Euglena gracilis*	LD_50_: 0.05 LD_100_: 0.09	LD_50_: n.t LD_100_: n.t	LD_50_: 0.8 LD_100_: 1.1	LD_50_: 0.5 LD_100_: 1.1	LD_50_: 0.6 LD_100_: 1.3	LD_50_: 0.5 LD_100_: 0.9	LD_50_: 0.5 LD_100_: 0.9	LD_50_: 0.5 LD_100_: 0.7	LD_50_: 0.1 LD_100_: 0.2	LD_50_: 0.5 LD_100_: 0.1	LD_50_: 0.2 LD_100_: 0.3
*Gregarina blattarum*	LD_50_: n.t LD_100_: n.t	LD_50_: 0.1 LD_100_: 0.3	LD_50_: 0.9 LD_100_: 1.1	LD_50_: 0.9 LD_100_: 1.0	LD_50_: 1.0 LD_100_: 1.1	LD_50_: 0.9 LD_100_: 1.0	LD_50_: 0.9 LD_100_: 1.0	LD_50_: 0.4 LD_100_: 0.7	LD_50_: 0.1 LD_100_: 0.4	LD_50_: 0.7 LD_100_: 0.3	LD_50_: 0.1 LD_100_: 0.2
*Amoeba proteus*	LD_50_: 0.07 LD_100_: 0.15	LD_50_: 0.3 LD_100_: 0.5	LD_50_: 0.8 LD_100_: 1.0	LD_50_: 0.6 LD_100_: 1.0	LD_50_: 0.9 LD_100_: 1.4	LD_50_: 0.5 LD_100_: 1.0	LD_50_: 0.5 LD_100_: 1.0	LD_50_: 0.5 LD_100_: 1.0	LD_50_: 0.1 LD_100_: 0.2	LD_50_: 0.5 LD_100_: 1.0	LD_50_: 0.1 LD_100_: 0.2
*Paramecium caudatum*	LD_50_: 0.001 LD_100_: 0.006	LD_50_: n.t LD_100_: n.t	LD_50_: 1.0 LD_100_: 1.3	LD_50_: 0.8 LD_100_: 1.2	LD_50_: 1.0 LD_100_: 1.5	LD_50_: 0.8 LD_100_: 1.2	LD_50_: 0.8 LD_100_: 1.2	LD_50_: 0.8 LD_100_: 1.2	LD_50_: 0.3 LD_100_: 0.5	LD_50_: 0.8 LD_100_: 1.2	LD_50_: 0.2 LD_100_: 0.3
*Pentatrichomonas hominis*	LD_50_: n.t LD_100_: n.t	LD_50_: 0.05 LD_100_: 0.14	LD_50_: 1.0 LD_100_: 1.5	LD_50_: 0.8 LD_100_: 1.0	LD_50_: 0.9 LD_100_: 1.3	LD_50_: 0.8 LD_100_: 1.0	LD_50_: 0.8 LD_100_: 1.0	LD_50_: 0.9 LD_100_: 1.1	LD_50_: 0.1 LD_100_: 0.3	LD_50_: 0.9 LD_100_: 1.1	LD_50_: 0.1 LD_100_: 0.2

^a^—chloramphenicol, ^b^—metronidazole, ^c^—in rate 1:1:1, ^d^—Manganese (II) chloride 10% solution, ^e^—Copper (II) carbonate hydroxide 10% solution, ^f^—Zinc carbonate 10% solution, ^g^—5% solution, n.t—not tested.

**Table 3 molecules-28-01395-t003:** LD_50_, LD_100_ values [%], for the tested mixtures of clove essential oil (*Syzygium aromaticum* (L.) Merr. and Perry) (S), organic acids (Acetic acid—A, Propionic acid—P, Lactic acid—L, Mixture of acids—M) and metal ion against selected protozoa.

Protozoa	Clove Essential Oil (*Syzygium aromaticum* (L.) Merr. and Perry)											
	Acetic Acid			Propionic Acid			Lactic Acid			Mixture of Acids ^a^		
	Cu ^b^	Mn ^c^	Zn ^d^	Cu ^b^	Mn ^c^	Zn ^d^	Cu ^b^	Mn ^c^	Zn ^d^	Cu ^b^	Mn ^c^	Zn ^d^
	SACu	SAMn	SAZn	SPCu	SPMn	SPZn	SLCu	SLMn	SLZn	SMCu	SMMn	SMZn
*Euglena gracilis*	LD_50_: 0.01 LD_100_:0.04	LD_50_: 0.02 LD_100_:0.03	LD_50_: 0.01 LD_100_:0.03	LD_50_: 0.02 LD_100_:0.03	LD_50_: 0.01 LD_100_:0.02	LD_50_: 0.01 LD_100_:0.02	LD_50_: 0.02 LD_100_:0.03	LD_50_: 0.01 LD_100_:0.03	LD_50_: 0.02 LD_100_:0.03	LD_50_: 0.001 LD_100_:0.003	LD_50_: 0.001 LD_100_:0.005	LD_50_: 0.001 LD_100_:0.002
*Gregarina blattarum*	LD_50_: 0.01 LD_100_:0.02	LD_50_: 0.01 LD_100_:0.015	LD_50_: 0.01 LD_100_:0.02	LD_50_: 0.01 LD_100_:0.02	LD_50_: 0.02 LD_100_:0.025	LD_50_: 0.01 LD_100_:0.02	LD_50_: 0.01 LD_100_:0.02	LD_50_: 0.01 LD_100_:0.02	LD_50_: 0.01 LD_100_:0.02	LD_50_: 0.002 LD_100_:0.004	LD_50_: 0.002 LD_100_:0.003	LD_50_: 0.002 LD_100_:0.005
*Amoeba proteus*	LD_50_: 0.01 LD_100_:0.02	LD_50_: 0.01 LD_100_:0.02	LD_50_: 0.01 LD_100_:0.015	LD_50_: 0.01 LD_100_:0.02	LD_50_: 0.01 LD_100_:0.02	LD_50_: 0.01 LD_100_:0.015	LD_50_: 0.01 LD_100_:0.02	LD_50_: 0.01 LD_100_:0.02	LD_50_: 0.01 LD_100_:0.02	LD_50_: 0.002 LD_100_:0.004	LD_50_: 0.001 LD_100_:0.002	LD_50_: 0.002 LD_100_:0.004
*Paramecium* *caudatum*	LD_50_: 0.01 LD_100_:0.03	LD_50_: 0.01 LD_100_:0.02	LD_50_: 0.02 LD_100_:0.025	LD_50_: 0.02 LD_100_:0.03	LD_50_: 0.02 LD_100_:0.03	LD_50_: 0.02 LD_100_:0.03	LD_50_: 0.01 LD_100_:0.03	LD_50_: 0.01 LD_100_:0.02	LD_50_: 0.01 LD_100_:0.02	LD_50_: 0.001 LD_100_:0.004	LD_50_: 0.002 LD_100_:0.005	LD_50_: 0.002 LD_100_:0.005
*Pentatrichomonas hominis*	LD_50_: 0.01 LD_100_:0.02	LD_50_: 0.01 LD_100_:0.02	LD_50_: 0.01 LD_100_:0.02	LD_50_: 0.01 LD_100_:0.02	LD_50_: 0.01 LD_100_:0.02	LD_50_: 0.01 LD_100_:0.02	LD_50_: 0.01 LD_100_:0.02	LD_50_: 0.01 LD_100_:0.02	LD_50_: 0.01 LD_100_:0.02	LD_50_: 0.004 LD_100_:0.006	LD_50_: 0.003 LD_100_:0.005	LD_50_: 0.003 LD_100_:0.004

^a^—in rate 1:1:1, ^b^—10% solution, ^c^—10% solution, ^d^—10% solution, SACu, SAMn, SAZn—Clove essential oil (S) with acetic acid (A) and Cu, Mn, Zn ions, respectively; SPCu, SPMn, SPZn—Clove essential oil (S) with propionic acid (P) and Cu, Mn, Zn ions, respectively; SLCu, SLMn, SLZn—Clove essential oil (S) with lactic acid (L) and Cu, Mn, Zn ions, respectively; SMCu, SMMn, SMZn—Clove essential oil (S) with mixture of acids (M) and Cu, Mn, Zn ions, respectively.

**Table 4 molecules-28-01395-t004:** LD_50_, LD_100_ values [%] of garlic essential oil (*Allium sativum* L.) and the components used in the study.

Protozoa	CH ^a^	M ^b^	Acetic Acid	Propionic Acid	Lactic Acid	Mixture of Acids ^c^	MnCl_2_ Solution ^d^	CH_2_Cu_2_O_5_ Solution ^e^	ZnCO_3_ Solution ^f^	Catalyst Solution ^g^	Garlic Essential Oil (*Allium sativum* L.)
*Euglena gracilis*	LD_50_: 0.05 LD_100_: 0.09	LD_50_: n.t LD_100_: n.t	LD_50_: 0.8 LD_100_: 1.1	LD_50_: 0.5 LD_100_: 1.1	LD_50_: 0.6 LD_100_: 1.3	LD_50_: 0.5 LD_100_: 0.9	LD_50_: 0.5 LD_100_: 0.7	LD_50_: 0.1 LD_100_: 0.2	LD_50_: 0.1 LD_100_: 0.3	LD_50_: 0.5 LD_100_: 0.1	LD_50_: 0.4 LD_100_: 0.7
*Gregarina blattarum*	LD_50_: n.t LD_100_: n.t	LD_50_: 0.1 LD_100_: 0.3	LD_50_: 0.9 LD_100_: 1.1	LD_50_: 0.9 LD_100_: 1.0	LD_50_: 1.0 LD_100_: 1.1	LD_50_: 0.9 LD_100_: 1.0	LD_50_: 0.4 LD_100_: 0.7	LD_50_: 0.1 LD_100_: 0.4	LD_50_: 0.2 LD_100_: 0.4	LD_50_: 0.7 LD_100_: 0.3	LD_50_: 0.3 LD_100_: 0.7
*Amoeba proteus*	LD_50_: 0.07 LD_100_: 0.15	LD_50_: 0.3 LD_100_: 0.5	LD_50_: 0.8 LD_100_: 1.0	LD_50_: 0.6 LD_100_: 1.0	LD_50_: 0.9 LD_100_: 1.4	LD_50_: 0.5 LD_100_: 1.0	LD_50_: 0.5 LD_100_: 1.0	LD_50_: 0.1 LD_100_: 0.2	LD_50_: 0.1 LD_100_: 0.2	LD_50_: 0.5 LD_100_: 1.0	LD_50_: 0.4 LD_100_: 0.6
*Paramecium caudatum*	LD_50_: 0.001 LD_100_: 0.006	LD_50_: n.t LD_100_: n.t	LD_50_: 1.0 LD_100_: 1.3	LD_50_: 0.8 LD_100_: 1.2	LD_50_: 1.0 LD_100_: 1.5	LD_50_: 0.8 LD_100_: 1.2	LD_50_: 0.8 LD_100_: 1.2	LD_50_: 0.3 LD_100_: 0.5	LD_50_: 0.3 LD_100_: 0.5	LD_50_: 0.8 LD_100_: 1.2	LD_50_: 0.5 LD_100_: 0.7
*Pentatrichomonas hominis*	LD_50_: n.t LD_100_: n.t	LD_50_: 0.05 LD_100_: 0.14	LD_50_: 1.0 LD_100_: 1.5	LD_50_: 0.8 LD_100_: 1.0	LD_50_: 0.9 LD_100_: 1.3	LD_50_: 0.8 LD_100_: 1.0	LD_50_: 0.9 LD_100_: 1.1	LD_50_: 0.1 LD_100_: 0.3	LD_50_: 0.2 LD_100_: 0.4	LD_50_: 0.9 LD_100_: 1.1	LD_50_: 0.6 LD_100_: 0.8

^a^—chloramphenicol, ^b^—metronidazole, ^c^—in rate 1:1:1, ^d^—Manganese (II) chloride 10% solution, ^e^—Copper (II) carbonate hydroxide 10% solution, ^f^—Zinc carbonate 10% solution, ^g^—5% solution, n.t—not tested.

**Table 5 molecules-28-01395-t005:** LD_50_, LD_100_ values [%] for the tested mixtures of garlic essential oil (*Allium sativum* L.) (G), organic acids (Acetic acid—A, Propionic acid—P, Lactic acid—L, Mixture of acids—M) and metal ion against selected protozoa.

Protozoa	Garlic Essential Oil (*Allium sativum* L.) (G)											
	Acetic Acid			Propionic Acid			Lactic Acid			Mixture of Acids ^a^		
	Cu ^b^	Mn ^c^	Zn ^d^	Cu ^b^	Mn ^c^	Zn ^d^	Cu ^b^	Mn ^c^	Zn ^d^	Cu ^b^	Mn ^c^	Zn ^d^
	GACu	GAMn	GAZn	GPCu	GPMn	GPZn	GLCu	GLMn	GLZn	GMCu	GMMn	GMZn
*Euglena gracilis*	LD_50_: 0.03 LD_100_:0.04	LD_50_: 0.02 LD_100_:0.05	LD_50_: 0.03 LD_100_:0.05	LD_50_: 0.02 LD_100_:0.06	LD_50_: 0.04 LD_100_:0.06	LD_50_: 0.05 LD_100_:0.06	LD_50_: 0.03 LD_100_:0.05	LD_50_: 0.02 LD_100_:0.04	LD_50_: 0.02 LD_100_:0.03	LD_50_:0.005 LD_100_:0.007	LD_50_:0.003 LD_100_:0.005	LD_50_:0.004 LD_100_:0.006
*Gregarina blattarum*	LD_50_: 0.01 LD_100_:0.05	LD_50_: 0.04 LD_100_:0.06	LD_50_: 0.03 LD_100_:0.06	LD_50_: 0.01 LD_100_:0.05	LD_50_: 0.02 LD_100_:0.04	LD_50_: 0.04 LD_100_:0.05	LD_50_: 0.01 LD_100_:0.04	LD_50_: 0.01 LD_100_:0.02	LD_50_: 0.01 LD_100_:0.03	LD_50_:0.003 LD_100_:0.006	LD_50_:0.003 LD_100_:0.005	LD_50_:0.003 LD_100_:0.006
*Amoeba proteus*	LD_50_: 0.01 LD_100_:0.05	LD_50_: 0.03 LD_100_:0.05	LD_50_: 0.03 LD_100_:0.04	LD_50_: 0.01 LD_100_:0.02	LD_50_: 0.03 LD_100_:0.05	LD_50_: 0.02 LD_100_:0.04	LD_50_: 0.01 LD_100_:0.03	LD_50_: 0.01 LD_100_:0.03	LD_50_: 0.01 LD_100_:0.04	LD_50_:0.004 LD_100_:0.005	LD_50_:0.001 LD_100_:0.003	LD_50_:0.004 LD_100_:0.006
*Paramecium* *caudatum*	LD_50_: 0.03 LD_100_:0.06	LD_50_: 0.02 LD_100_:0.03	LD_50_: 0.03 LD_100_:0.04	LD_50_: 0.04 LD_100_:0.05	LD_50_: 0.02 LD_100_:0.06	LD_50_: 0.02 LD_100_:0.04	LD_50_: 0.01 LD_100_:0.03	LD_50_: 0.01 LD_100_:0.04	LD_50_: 0.01 LD_100_:0.02	LD_50_:0.004 LD_100_:0.006	LD_50_:0.002 LD_100_:0.004	LD_50_:0.004 LD_100_:0.007
*Pentatrichomonas hominis*	LD_50_: 0.02 LD_100_:0.04	LD_50_: 0.04 LD_100_:0.05	LD_50_: 0.04 LD_100_:0.05	LD_50_: 0.03 LD_100_:0.065	LD_50_: 0.04 LD_100_:0.06	LD_50_: 0.04 LD_100_:0.07	LD_50_: 0.02 LD_100_:0.06	LD_50_: 0.01 LD_100_:0.05	LD_50_: 0.03 LD_100_:0.05	LD_50_:0.03 LD_100_:0.06	LD_50_:0.03 LD_100_:0.05	LD_50_:0.003 LD_100_:0.004

^a^—in rate 1:1:1, ^b^—10% solution, ^c^—10% solution, ^d^—10% solution, GACu, GAMn, GAZn—Garlic essential oil (G) with acetic acid (A) and Cu, Mn, Zn ions, respectively; GPCu, GPMn, GPZn—Garlic essential oil (G) with propionic acid (P) and Cu, Mn, Zn ions, respectively; GLCu, GLMn, GLZn—Garlic essential oil (G) with lactic acid (L) and Cu, Mn, Zn ions, respectively; GMCu, GMMn, GMZn—Garlic essential oil (G) with mixture of acids (M) and Cu, Mn, Zn ions, respectively.

**Table 6 molecules-28-01395-t006:** LD_50_, LD_100_ values [%] of Ceylon cinnamon essential oil (*Cinnamomum verum* J. Presl) and the components used in the study.

Protozoa	CH ^a^	M ^b^	Acetic Acid	Propionic Acid	Lactic Acid	Mixture of Acids ^c^	MnCl_2_ Solution ^d^	CH_2_Cu_2_O_5_ Solution ^e^	ZnCO_3_ Solution ^f^	Catalyst Solution ^g^	Ceylon Cinnamon Essential Oil (*Cinnamomum verum* J. Presl)
*Euglena gracilis*	LD_50_: 0.05 LD_100_: 0.09	LD_50_: n.t LD_100_: n.t	LD_50_: 0.8 LD_100_: 1.1	LD_50_: 0.5 LD_100_: 1.1	LD_50_: 0.6 LD_100_: 1.3	LD_50_: 0.5 LD_100_: 0.9	LD_50_: 0.5 LD_100_: 0.7	LD_50_: 0.1 LD_100_: 0.2	LD_50_: 0.1 LD_100_: 0.3	LD_50_: 0.5 LD_100_: 0.1	LD_50_: 0.2 LD_100_: 0.3
*Gregarina blattarum*	LD_50_: n.t LD_100_: n.t	LD_50_: 0.1 LD_100_: 0.3	LD_50_: 0.9 LD_100_: 1.1	LD_50_: 0.9 LD_100_: 1.0	LD_50_: 1.0 LD_100_: 1.1	LD_50_: 0.9 LD_100_: 1.0	LD_50_: 0.4 LD_100_: 0.7	LD_50_: 0.1 LD_100_: 0.4	LD_50_: 0.2 LD_100_: 0.4	LD_50_: 0.7 LD_100_: 0.3	LD_50_: 0.1 LD_100_: 0.35
*Amoeba proteus*	LD_50_: 0.07 LD_100_: 0.15	LD_50_: 0.3 LD_100_: 0.5	LD_50_: 0.8 LD_100_: 1.0	LD_50_: 0.6 LD_100_: 1.0	LD_50_: 0.9 LD_100_: 1.4	LD_50_: 0.5 LD_100_: 1.0	LD_50_: 0.5 LD_100_: 1.0	LD_50_: 0.1 LD_100_: 0.2	LD_50_: 0.1 LD_100_: 0.2	LD_50_: 0.5 LD_100_: 1.0	LD_50_: 0.5 LD_100_: 0.6
*Paramecium caudatum*	LD_50_: 0.001 LD_100_: 0.006	LD_50_: n.t LD_100_: n.t	LD_50_: 1.0 LD_100_: 1.3	LD_50_: 0.8 LD_100_: 1.2	LD_50_: 1.0 LD_100_: 1.5	LD_50_: 0.8 LD_100_: 1.2	LD_50_: 0.8 LD_100_: 1.2	LD_50_: 0.3 LD_100_: 0.5	LD_50_: 0.3 LD_100_: 0.5	LD_50_: 0.8 LD_100_: 1.2	LD_50_: 0.2 LD_100_: 0.45
*Pentatrichomonas hominis*	LD_50_: n.t LD_100_: n.t	LD_50_: 0.05 LD_100_: 0.14	LD_50_: 1.0 LD_100_: 1.5	LD_50_: 0.8 LD_100_: 1.0	LD_50_: 0.9 LD_100_: 1.3	LD_50_: 0.8 LD_100_: 1.0	LD_50_: 0.9 LD_100_: 1.1	LD_50_: 0.1 LD_100_: 0.3	LD_50_: 0.2 LD_100_: 0.4	LD_50_: 0.9 LD_100_: 1.1	LD_50_: 0.2 LD_100_: 0.7

^a^—chloramphenicol, ^b^—metronidazole, ^c^—in rate 1:1:1, ^d^—Manganese (II) chloride 10% solution, ^e^—Copper (II) carbonate hydroxide 10% solution, ^f^—Zinc carbonate 10% solution, ^g^—5% solution, n.t—not tested.

**Table 7 molecules-28-01395-t007:** LD_50_, LD_100_ values [%] for the tested mixtures of Ceylon cinnamon essential oil (*Cinnamomum verum* J. Presl) (C), organic acids (Acetic acid—A, Propionic acid—P, Lactic acid—L, Mixture of acids—M) and metal ion against selected protozoa.

Protozoa	Ceylon Cinnamon Essential Oil (*Cinnamomum verum* J. Presl) (C)											
	Acetic Acid			Propionic Acid			Lactic Acid			Mixture of Acids ^a^		
	Cu ^b^	Mn ^c^	Zn ^d^	Cu ^b^	Mn ^c^	Zn ^d^	Cu ^b^	Mn ^c^	Zn ^d^	Cu ^b^	Mn ^c^	Zn ^d^
	CACu	CAMn	CAZn	CPCu	CPMn	CPZn	CLCu	CLMn	CLZn	CMCu	CMMn	CMZn
*Euglena gracilis*	LD_50_: 0.02 LD_100_:0.03	LD_50_: 0.01 LD_100_:0.02	LD_50_: 0.01 LD_100_:0.02	LD_50_: 0.01 LD_100_:0.02	LD_50_: 0.01 LD_100_:0.02	LD_50_: 0.01 LD_100_:0.02	LD_50_: 0.02 LD_100_:0.03	LD_50_: 0.01 LD_100_:0.02	LD_50_: 0.01 LD_100_:0.02	LD_50_:0.005 LD_100_:0.007	LD_50_:0.001 LD_100_:0.003	LD_50_:0.003 LD_100_:0.006
*Gregarina blattarum*	LD_50_: 0.02 LD_100_:0.04	LD_50_: 0.02 LD_100_:0.03	LD_50_: 0.02 LD_100_:0.03	LD_50_: 0.02 LD_100_:0.035	LD_50_: 0.02 LD_100_:0.03	LD_50_: 0.01 LD_100_:0.03	LD_50_: 0.02 LD_100_:0.03	LD_50_: 0.01 LD_100_:0.02	LD_50_: 0.01 LD_100_:0.02	LD_50_:0.005 LD_100_:0.006	LD_50_:0.003 LD_100_:0.006	LD_50_:0.003 LD_100_:0.005
*Amoeba proteus*	LD_50_: 0.04 LD_100_:0.055	LD_50_: 0.02 LD_100_:0.05	LD_50_: 0.04 LD_100_:0.05	LD_50_: 0.05 LD_100_:0.06	LD_50_: 0.03 LD_100_:0.05	LD_50_: 0.05 LD_100_:0.06	LD_50_: 0.04 LD_100_:0.05	LD_50_: 0.02 LD_100_:0.04	LD_50_: 0.02 LD_100_:0.04	LD_50_:0.003 LD_100_:0.005	LD_50_:0.001 LD_100_:0.003	LD_50_:0.004 LD_100_:0.006
*Paramecium* *caudatum*	LD_50_: 0.03 LD_100_:0.05	LD_50_: 0.03 LD_100_:0.04	LD_50_: 0.03 LD_100_:0.04	LD_50_: 0.03 LD_100_:0.04	LD_50_: 0.02 LD_100_:0.045	LD_50_: 0.02 LD_100_:0.04	LD_50_: 0.02 LD_100_:0.03	LD_50_: 0.02 LD_100_:0.03	LD_50_: 0.01 LD_100_:0.02	LD_50_:0.002 LD_100_:0.004	LD_50_:0.002 LD_100_:0.006	LD_50_:0.004 LD_100_:0.007
*Pentatrichomonas hominis*	LD_50_: 0.06 LD_100_:0.07	LD_50_: 0.05 LD_100_:0.065	LD_50_: 0.04 LD_100_:0.06	LD_50_: 0.03 LD_100_:0.055	LD_50_: 0.04 LD_100_:0.05	LD_50_: 0.05 LD_100_:0.07	LD_50_: 0.04 LD_100_:0.05	LD_50_: 0.04 LD_100_:0.05	LD_50_: 0.04 LD_100_:0.05	LD_50_:0.05 LD_100_:0.06	LD_50_:0.03 LD_100_:0.05	LD_50_:0.04 LD_100_:0.055

^a^—in rate 1:1:1, ^b^—10% solution, ^c^—10% solution, ^d^—10% solution, CACu, CAMn, CAZn—Ceylon cinnamon essential oil (C) with acetic acid (A) and Cu, Mn, Zn ions, respectively; CPCu, CPMn, CPZn—Ceylon cinnamon essential oil (C) with propionic acid (P) and Cu, Mn, Zn ions, respectively; CLCu, CLMn, CLZn—Ceylon cinnamon essential oil (C) with lactic acid (L) and Cu, Mn, Zn ions, respectively; CMCu, CMMn, CMZn—Ceylon cinnamon essential oil (C) with mixture of acids (M) and Cu, Mn, Zn ions, respectively.

**Table 8 molecules-28-01395-t008:** LD_50_, LD_100_ values [%] of rosemary essential oil (*Rosmarinus officinalis* L.) and the components used in the study.

Protozoa	CH ^a^	M ^b^	Acetic Acid	Propionic Acid	Lactic Acid	Mixture of Acids ^c^	MnCl_2_ Solution ^d^	CH_2_Cu_2_O_5_ Solution ^e^	ZnCO_3_ Solution ^f^	Catalyst Solution ^g^	Rosemary Essential Oil (*Rosmarinus officinalis* L.)
*Euglena gracilis*	LD_50_: 0.05 LD_100_: 0.09	LD_50_: n.t LD_100_: n.t	LD_50_: 0.8 LD_100_: 1.1	LD_50_: 0.5 LD_100_: 1.1	LD_50_: 0.6 LD_100_: 1.3	LD_50_: 0.5 LD_100_: 0.9	LD_50_: 0.5 LD_100_: 0.7	LD_50_: 0.1 LD_100_: 0.2	LD_50_: 0.1 LD_100_: 0.3	LD_50_: 0.5 LD_100_: 0.1	LD_50_: 0.2 LD_100_: 0.6
*Gregarina blattarum*	LD_50_: n.t LD_100_: n.t	LD_50_: 0.1 LD_100_: 0.3	LD_50_: 0.9 LD_100_: 1.1	LD_50_: 0.9 LD_100_: 1.0	LD_50_: 1.0 LD_100_: 1.1	LD_50_: 0.9 LD_100_: 1.0	LD_50_: 0.4 LD_100_: 0.7	LD_50_: 0.1 LD_100_: 0.4	LD_50_: 0.2 LD_100_: 0.4	LD_50_: 0.7 LD_100_: 0.3	LD_50_: 0.2 LD_100_: 0.4
*Amoeba proteus*	LD_50_: 0.07 LD_100_: 0.15	LD_50_: 0.3 LD_100_: 0.5	LD_50_: 0.8 LD_100_: 1.0	LD_50_: 0.6 LD_100_: 1.0	LD_50_: 0.9 LD_100_: 1.4	LD_50_: 0.5 LD_100_: 1.0	LD_50_: 0.5 LD_100_: 1.0	LD_50_: 0.1 LD_100_: 0.2	LD_50_: 0.1 LD_100_: 0.2	LD_50_: 0.5 LD_100_: 1.0	LD_50_: 0.3 LD_100_: 0.5
*Paramecium caudatum*	LD_50_: 0.001 LD_100_: 0.006	LD_50_: n.t LD_100_: n.t	LD_50_: 1.0 LD_100_: 1.3	LD_50_: 0.8 LD_100_: 1.2	LD_50_: 1.0 LD_100_: 1.5	LD_50_: 0.8 LD_100_: 1.2	LD_50_: 0.8 LD_100_: 1.2	LD_50_: 0.3 LD_100_: 0.5	LD_50_: 0.3 LD_100_: 0.5	LD_50_: 0.8 LD_100_: 1.2	LD_50_: 0.1 LD_100_: 0.5
*Pentatrichomonas hominis*	LD_50_: n.t LD_100_: n.t	LD_50_: 0.05 LD_100_: 0.14	LD_50_: 1.0 LD_100_: 1.5	LD_50_: 0.8 LD_100_: 1.0	LD_50_: 0.9 LD_100_: 1.3	LD_50_: 0.8 LD_100_: 1.0	LD_50_: 0.9 LD_100_: 1.1	LD_50_: 0.1 LD_100_: 0.3	LD_50_: 0.2 LD_100_: 0.4	LD_50_: 0.9 LD_100_: 1.1	LD_50_: 0.3 LD_100_: 0.55

^a^—chloramphenicol, ^b^—metronidazole, ^c^—in rate 1:1:1, ^d^—Manganese (II) chloride 10% solution, ^e^—Copper (II) carbonate hydroxide 10% solution, ^f^—Zinc carbonate 10% solution, ^g^—5% solution, n.t—not tested.

**Table 9 molecules-28-01395-t009:** LD_50_, LD_100_ values [%] for the tested mixtures of rosemary essential oil (*Rosmarinus officinalis* L.), organic acids (Acetic acid—A, Propionic acid—P, Lactic acid—L, Mixture of acids—M) and metal ion against selected protozoa.

Protozoa	Rosemary Essential Oil (*Rosmarinus officinalis* L.)											
	Acetic Acid			Propionic Acid			Lactic Acid			Mixture of Acids ^a^		
	Cu ^b^	Mn ^c^	Zn ^d^	Cu ^b^	Mn ^c^	Zn ^d^	Cu ^b^	Mn ^c^	Zn ^d^	Cu ^b^	Mn ^c^	Zn ^d^
	RACu	RAMn	RAZn	RPCu	RPMn	RPZn	RLCu	RLMn	RLZn	RMCu	RMMn	RMZn
*Euglena gracilis*	LD_50_: 0.03 LD_100_:0.05	LD_50_: 0.02 LD_100_:0.05	LD_50_: 0.03 LD_100_:0.06	LD_50_: 0.03 LD_100_:0.05	LD_50_: 0.02 LD_100_:0.045	LD_50_: 0.04 LD_100_:0.05	LD_50_: 0.03 LD_100_:0.05	LD_50_: 0.02 LD_100_:0.04	LD_50_: 0.02 LD_100_:0.03	LD_50_: 0.002 LD_100_:0.005	LD_50_: 0.001 LD_100_:0.004	LD_50_: 0.002 LD_100_:0.004
*Gregarina blattarum*	LD_50_: 0.02 LD_100_:0.04	LD_50_: 0.02 LD_100_:0.03	LD_50_: 0.02 LD_100_:0.03	LD_50_: 0.02 LD_100_:0.04	LD_50_: 0.02 LD_100_:0.04	LD_50_: 0.03 LD_100_:0.05	LD_50_: 0.01 LD_100_:0.03	LD_50_: 0.02 LD_100_:0.03	LD_50_: 0.01 LD_100_:0.02	LD_50_: 0.002 LD_100_:0.004	LD_50_: 0.002 LD_100_:0.003	LD_50_: 0.003 LD_100_:0.005
*Amoeba proteus*	LD_50_: 0.04 LD_100_:0.05	LD_50_: 0.02 LD_100_:0.03	LD_50_: 0.04 LD_100_:0.05	LD_50_: 0.03 LD_100_:0.04	LD_50_: 0.02 LD_100_:0.05	LD_50_: 0.02 LD_100_:0.03	LD_50_: 0.03 LD_100_:0.04	LD_50_: 0.02 LD_100_:0.04	LD_50_: 0.04 LD_100_:0.05	LD_50_: 0.002 LD_100_:0.004	LD_50_: 0.002 LD_100_:0.005	LD_50_: 0.003 LD_100_:0.004
*Paramecium* *caudatum*	LD_50_: 0.03 LD_100_:0.04	LD_50_: 0.03 LD_100_:0.04	LD_50_: 0.03 LD_100_:0.04	LD_50_: 0.03 LD_100_:0.05	LD_50_: 0.03 LD_100_:0.04	LD_50_: 0.03 LD_100_:0.05	LD_50_: 0.01 LD_100_:0.04	LD_50_: 0.02 LD_100_:0.03	LD_50_: 0.02 LD_100_:0.03	LD_50_: 0.001 LD_100_:0.004	LD_50_: 0.002 LD_100_:0.005	LD_50_: 0.002 LD_100_:0.005
*Pentatrichomonas hominis*	LD_50_: 0.03 LD_100_:0.05	LD_50_: 0.05 LD_100_:0.055	LD_50_: 0.04 LD_100_:0.06	LD_50_: 0.03 LD_100_:0.04	LD_50_: 0.04 LD_100_:0.05	LD_50_: 0.03 LD_100_:0.04	LD_50_: 0.03 LD_100_:0.05	LD_50_: 0.04 LD_100_:0.05	LD_50_: 0.03 LD_100_:0.04	LD_50_: 0.007 LD_100_:0.009	LD_50_: 0.006 LD_100_:0.008	LD_50_: 0.005 LD_100_:0.008

^a^—in rate 1:1:1, ^b^—10% solution, ^c^—10% solution, ^d^—10% solution, RACu, RAMn, RAZn—Rosemary essential oil (R) with acetic acid (A) and Cu, Mn, Zn ions, respectively; RPCu, RPMn, RPZn—Rosemary essential oil (R) with propionic acid (P) and Cu, Mn, Zn ions, respectively; RLCu, RLMn, RLZn—Rosemary essential oil (R) with lactic acid (L) and Cu, Mn, Zn ions, respectively; RMCu, RMMn, RMZn—Rosemary essential oil (R) with mixture of acids (M) and Cu, Mn, Zn ions, respectively.

**Table 15 molecules-28-01395-t015:** IC_50_ values against normal and tumour lines incubated for 48 h with essential oil of Ceylon cinnamon (*Cinnamomum verum* J. Presl) and the components used in the study.

Cell Lines	Mixture of Acids	+MnCl_2_	+CH_2_Cu_2_O_5_	+ZnCO_3_
NHDF	1.43 (±0.16)	0.54 (±0.2)	19.28 (±1.3)	0.73 (±0.16)
A549	0.12 (±0.03)	0.10 (±0.01)	0.15 (±0.02)	0.07 (±0.01)
MCF7	0.34 (±0.09)	0.70 (±0.09)	0.67 (±0.15)	0.15 (±0.03)
LOVO	0.14 (±0.03)	0.06 (±0.01)	0.09 (±0.01)	0.07 (±0.01)
HT29	1.43 (±0.16)	0.54 (±0.2)	19.28 (±1.3)	0.73 (±0.16)

**Table 16 molecules-28-01395-t016:** IC_50_ values against normal and tumour lines incubated for 48 h with garlic essential oil (*Allium sativum* L.) and the components used in the study.

Cell Lines	Mixture of Acids	+MnCl_2_	+CH_2_Cu_2_O_5_	+ZnCO_3_
NHDF	0.62 (±0.11)	0.91 (±0.07)	0.92 (±0.13)	1.17 (±0.14)
A549	0.40 (±0.07)	0.46 (±0.)	NA	NA
MCF7	0.13 (±0.03)	0.07 (±0.)	0.13 (±0.03)	0.11 (±0.01)
LOVO	NA	0.39 (±0.)	0.31 (±0.15)	NA
HT29	0.46 (±0.12)	0.06 (±0.01)	NA	NA

NA—not active.

**Table 17 molecules-28-01395-t017:** IC_50_ values against normal and tumour lines incubated for 48 h with clove essential oil (*Syzygium aromaticum* (L.) Merr. and Perry) and the components used in the study.

Cell Lines	Mixture of Acids	+MnCl_2_	+CH_2_Cu_2_O_5_	+ZnCO_3_
NHDF	0.51 (±0.22)	0.71 (±0.12)	0.38 (±0.12)	0.71 (±0.21)
A549	0.38 (±0.10)	0.22 (±0.02)	NA	0.13 (±0.03)
MCF7	0.22 (±0.06)	0.13 (±0.02)	NA	0.05 (±0.01)
LOVO	0.37 (±0.01)	0.13 (±0.03)	0.08 (±0.02)	0.05 (±0.01)
HT29	NA	NA	NA	0.15 (±0.03)

NA—not active.

**Table 18 molecules-28-01395-t018:** IC50 values against normal and tumour lines incubated for 48 h with rosemary essential oil (*Rosmarinus officinalis*) and the components used in the study.

Cell Lines	Mixture of Acids	+MnCl_2_	+CH_2_Cu_2_O_5_	+ZnCO_3_
NHDF	1.28 (±0.21)	1.51 (±0.31)	1.24 (±0.21)	NA
A549	NA	0.15 (±0.03)	NA	0.07 (±0.02)
MCF7	NA	0.11 (±0.01)	0.35 (±0.03)	NA
LOVO	NA	NA	NA	0.07 (±0.01)
HT29	NA	NA	NA	NA

NA—not active.

## Data Availability

The data presented in this study are available in the tables and Appendix A.

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
