# Peer review of "In Vitro Evaluation of Antiprotozoal Properties, Cytotoxicity Effect and Anticancer Activity of New Essential-Oil Based Phytoncide Mixtures"

_molecules, 2023, doi:10.3390/molecules28031395_

Round 1

Reviewer 1 Report

In this research article entitled " In vitro evaluation of antiprotozoal properties and cytotoxicity of new essential-oil based phytoncide mixtures " the authors explored the in vitro antiprotozoal activity of selected 4 essential oils alone and in mixtures with organic acids (acetic, propionic, lactic) and metal ions Cu, Mn, Zn) against five selected model protozoa (Euglena gracilis, Gregarina blattarum, Amoeba proteus, Paramecium caudatum, Pentatrichomonas hominis). Moreover, authors tested the cytotoxicity and anticancer activity of the EOs combinations on the human fibroblasts (NHDF) and humane cancer cell lines (A549, MCF7, LoVo, HT29). Quantitatively, there were performed enough experiments and results and discussion were presented and analyzed well in most cases. Tables and figures are mainly clear and organized. However, I mention below some points that should be considered before processing further.

- Line 116 – 122 – in numeric data please replace the commas with dots

- Table 2, 3, 4, 5, 6, 7, 8 and 9 - it would be more appropriate to state that the data are given in percentages (%) directly in the name of the table and then there is no need to state a separate % for each numerical data in the tables

- I recommend shortening the number of tables in the manuscript. It would be more appropriate to list all the essential oils and their chemical composition (Table 10, 11, 13 and 14) in one complete table (some components are repeated) and it is unnecessary to have so many tables in the article.

- please explain the abbreviation TIC in the legend of tables 10,11,13 and 14

- Line 314 – in numeric data please replace the commas with dots

- Line 339 – in numeric data please replace the commas with dots

- section Material and Methods - please state the origin of the standards used in the GC-MS analysis.

In conclusion, the article is well written, a few adjustments need to be made in it, and after minor corrections it could be published in Molecules. However, I suggest that the title of the article should be changed and the testing of the cytotoxic effect and anticancer activity of EOs on human fibroblasts and cancer cell lines should also be included, in my opinion, the article would have a much higher impact and viewership.

Reviewer 2 Report

The study done by Iwiński et al. Mixtures of essential oils clove, garlic, Ceylon cinnamon, and rosemary with organic acids (acetic, propionic, lactic) and metal ions (Cu, Mn, Zn) were tested against five selected model protozoa. The cytotoxicity and a potential anticancer activity of the combinations were tested on the human fibroblasts (NHDF) and humane cancer cell lines (A549, MCF7, LoVo, HT29). According to the results, author claimed that all the mixtures showed very good antiprotozoal properties. The most efficient were combinations of clove and rosemary essential oils, mixtures of acids, and Mn ions. All mixtures tested did not show cytotoxicity against normal cells but show growth inhibition against cancer cell lines. The study is well written, data are clearly represented, and results support their findings.

 Some minor comments needs to be addressed prior acceptance: 

1.     Line 110: Replace “antibacterial activity” with “antiprotozoal activity”

2.     Line 279: Remove table 15-foot note “NA—not active”, keep it in the rest (16-18)

3.     Line 402: Replace “tree” with “three”

4.     Line 471: Replace (1*104 cells per well) to (1* 104 cells per well)

5.     Line 473: Add concentration of TCA used (It is usually 50%)

6.     I would prefer to mention the positive control used in cytotoxicity experiment

7.     For authors, to take into consideration, regarding cytotoxicity experiments in the future, I used to leave peripheral wells in the 96 well microtiter plate (36 wells at edges) filled with only buffer as dam, as these wells are more likely subjected to evaporation upon incubation, hence affecting reading results.

Reviewer 3 Report

General comments

·       I don't quite understand what antiprotozoal research has in common with cytotoxic/anticancer effect on cell lines? After all, they are 2 completely different activities. Please detail the purpose of the research.

·       In the introduction, the authors only gently raise the issue of diseases caused by protozoa, but do not write what these diseases are, etc.

·       Emphasise the novelty of the study, please.

·       Why did the authors choose such and not other cell line for their study? Please provide justification.

·       Maybe to keywords “anticancer” should be added…

·       Discuss the limitations of the study, please.

Detailed comments

·       Line 23: One parenthesis is missing - the opening one. Add please.

·       Line 26: combinations of what?

·       Line 26: correct “humane” to “human”.

·       Line 31: “Nevertheless” write with a small letter.

·       Line 52: Plasmodium falciparum – should be written with italics.

·       Line 57: One parenthesis is missing.

·       In Table 1 there is no need to repeat the “essential oil” in all rows of the table in the first column from the left.

·       Lines 253-276: please adjust text.

·       All abbreviations should be defined when used for the first time.

·       There are several typhos in the manuscript.

·       What was the viability of particular cell lines taken to the experiment? How was it measured?

Round 2

Reviewer 3 Report

I have no more comments.